# Nature Tourism on the Colombian—Ecuadorian Amazonian Border: History, Current Situation, and Challenges

Carlos Mestanza-Ramón [1,2,3,*] and José Luis Jiménez-Caballero [1]

1　Departamento Economía Financiera y Dirección de Operaciones, Universidad de Sevilla, 41018 Sevilla, Spain; jjimenez@us.es
2　Instituto Superior Tecnológico Universitario Oriente, La Joya de los Sachas 220101, Ecuador
3　Research Group YASUNI-SDC, Escuela Superior Politécnica de Chimborazo, Sede Orellana, El Coca EC 220001, Ecuador
*　Correspondence: cmestanza@ug.uchile.cl

**Abstract:** Global conflicts can severely affect a nation's tourism activities. Tourism can also be seriously affected by health problems such as epidemics or pandemics. It is important to establish strategies to be prepared for adverse situations. The objective of this study focused on analyzing nature tourism from a post-conflict and post-COVID-19 situation in the Amazonian border of Colombia (Department of Putumayo) and Ecuador (Province of Sucumbíos), which will contribute to establishing future strategic management scenarios. In order to respond to this objective, a systematic bibliographic review was carried out, accompanied by fieldwork (interviews). The results indicate that in the face of adverse situations, the tourism industry has the capacity to be resilient. The success of its recovery will be directly proportional to its capacity to create policies and strategies that allow it to take advantage of natural resources and turn them into an opportunity for the socioeconomic development of its population.

**Keywords:** post-conflict; post-pandemic; Sucumbíos; Putumayo; resilience; COVID-19; sustainable tourism

## 1. Introduction

Tourism provides livelihoods for millions of people and allows them to appreciate their own and other cultures and nature. The tourism industry grew rapidly in recent years, becoming the world's third largest export category (after fuels and chemicals) [1,2]. In 2019, international tourist arrivals reached 1.5 billion thanks to a decade of uninterrupted growth, and accounted for 10.3% of global GDP, generating 330 million jobs. By 2020, with the outbreak of COVID-19, tourism activities were forced to come to an abrupt halt and enter a deep economic crisis [3,4]. Due to global border closures, tourism has been one of the sectors most affected by the coronavirus pandemic [5]. The crisis is unprecedented in size and scope, and has affected the livelihoods of millions of people around the world [6,7]. In this period, USD 320 billion in tourism exports were lost, more than triple the losses during the global financial crisis of 2009 [8]. This situation offers an opportunity to rethink the tourism sector and promote new opportunities for nature destinations and other sites neglected by conflicts or social problems.

Nature tourism is the segment of tourism directly related to the sustainable use of natural resources. The main motivation of this segment is to carry out outdoor activities in contact with nature and its biodiversity [9–13]. This type of tourism provides the receiving communities with the means to improve their quality of life, seeking to encourage sustainable development. This tourism sector can be grouped according to the tourist's interest in four modalities: adventure tourism, ecotourism, geotourism, and wildlife tourism (flora and fauna) [14]. Nature tourism was projected to be one of the fastest growing types of tourism in the world, with rates between 25% and 30% per year until before COVID-19 [15]. Nature tourism is directly related to development and sustainable tourism, and is

practically related to natural habitats and their biodiversity. Sustainable tourism describes policies, practices, and programs that consider not only the expectations of tourists in terms of responsible management of natural resources, but also the needs of the communities that support or are affected by tourism projects and the environment [2,16,17].

Security problems affect 60% of the countries which are highly dependent on tourism. Aspects such as personal safety, cost, travel time, comfort, and flexibility influence travelers' decisions when choosing a destination for vacationing [18,19]. Regarding personal safety, it has been demonstrated in the literature that the reliability of police services is the sixth most relevant indicator distinguishing developed from developing destinations [20]. Incidents of terrorism, armed conflicts, crime, natural disasters, political instability, and epidemic outbreaks have a negative impact on the image of a destination and represent a major challenge when trying to strengthen tourism and create new life opportunities [16,21]. It is important to differentiate that disasters or minor short-term problems do not significantly affect a destination, unlike historical conflicts which are unlikely to disappear and are totally incompatible with the development of tourism in a region [17]. Tourism is sometimes used as an opportunity to re-establish relationships in places that have historically been scarred by conflict [22]. Post-conflict areas that are politically unstable as a result of war have a negative impact on tourism, affecting revenues and visitor numbers. These areas take time for tourists to take the risk of visiting and sometimes never fully recover from the negative perceptions of violence, insecurity, and instability [23]. On the other hand, cases have been observed in which, by implementing strategies, the image of being devastated by conflicts has been quickly cleaned up. In certain cases, war stories and their ruins can serve as a resource for groups interested in cultural and heritage tourism [24,25].

Conflict is a reality present in all strata and professional spheres of our society, and, on most occasions, it is of a multiple nature. The analysis of conflict situations affecting the tourism industry is currently an area of wide debate and reflection due to its relevant impact on business strategy and management [26]. The tourism industry is complex, involving multiple agents with different interests. As a result, risk scenarios are created that generate uncertainty both for the achievement of the organizational objectives of the tourism agents and for the success of the destination [27]. On the other hand, tourism planning and management have made progress, but still leave much to be desired in terms of the applicability of action strategies. A large percentage of mistakes in tourism management are due to the adoption of inappropriate measures [28].

The Colombian–Ecuadorian Amazon covers an area of 7.4 million km$^2$, which represents 5% of the world's continental surface. It is estimated that the Amazon forests offer the greatest biodiversity of flora and fauna on the planet, with 50% of the total number of existing living species cohabiting on its surface. It is a geographic region of natural, social, and cultural diversity thanks to the legacy of its ancestors and ancestral peoples. Historically, this border region has been characterized by security problems as a result of the armed conflict between the Colombian Revolutionary Armed Forces (FARC) and the government. It is an eminently jungle-like region, with variations in relief and a wide variety of geographic features. It has an enormous variety of natural landscapes, with the hydrographic component standing out as the most used and exploited resource for commerce, consumption, and recreation [26,27].

Tourism has been characterized by its vulnerability to crises and disasters (e.g., the Fukushima nuclear disaster in Japan, the September 11 terrorist attacks in the United States, the 2004 tsunami in several Asian countries, and the current coronavirus pandemic) [29–31]. Since these events, there has been a growing body of literature and knowledge regarding the preparedness and response to adverse situations in tourism [32]. However, there is a need to focus efforts and strengthen new research related to conflict, political instability, health issues, and tourism [33]. In this sense, the purpose of this study was to analyze the nature tourism sector in a post-conflict and post-COVID situation in the Amazonian border of Colombia (Department of Putumayo) and Ecuador (Province of Sucumbíos). To achieve the proposed objective, a bibliographic review and interviews were carried

out, and the information was analyzed by means of expert judgment. In order to fulfill the proposed objective, it was necessary to answer the following questions: (i) How did tourism develop in an era of armed conflict?; (ii) What is the current situation of tourism in a COVID-19 pandemic situation; and (iii) What are the strategic management scenarios for nature tourism in the Colombian–Ecuadorian Amazon region in a post-conflict and post-pandemic situation?

## 2. Methodology

The methodology was divided into three sections. The first focuses on the analysis and description of the historical situation of the armed conflict and post-conflict in the Amazonian border of Colombia (Department of Putumayo) and Ecuador (Province of Sucumbíos) with respect to tourism. The second section analyzes the current tourism situation focusing on the COVID-19 pandemic crisis and its effects. Finally, the challenges of tourism in a post-conflict and post-pandemic situation are discussed.

To respond to the first and second sections, a systematic literature review was conducted in which the content of the documents selected for analysis was reviewed. In the search process, articles published from 2016 onwards were considered in high-impact databases such as Scopus and Web of Science. Search filters were applied to the title, keywords and abstract (Table 1, resulting in 31 documents. To improve the selection of bibliographic material, a second manual filter of complete reading of the titles and abstracts was performed on the documents returned after the search. Seven documents were obtained on the conflict and post-conflict situation of tourism; eight documents focused on the current situation of COVID-19 and tourism in the area of influence. Additionally, a semi-structured interview was conducted with 100% of the foreign tourists (125 people) (except Ecuadorian and Colombian tourists) who crossed the San Miguel migration control point, on the Amazonian border of Colombia and Ecuador, between January and February, 2021. The questions focused on what their main fears were before visiting the transit zone.

In addition to the analysis of the bibliographic resources of the high-impact database (Table 1), a review of the gray literature was complemented, corresponding to a set of documents of various types which have not undergone review or editing processes and are available in non-conventional channels. The documents selected were reviews of land management plans, tourism development plans, NGO reports, environmental management plans, tourism brochures and news. This important information helped to identify the historical and current situation of the study area.

Finally, the information obtained from the bibliographic analysis (Table 1) and the gray literature was used to establish future strategic management scenarios for nature tourism in the post-conflict and post-combat situation in the Amazonian border of Colombia (Department of Putumayo) and Ecuador (Province of Sucumbíos). This information served as the basis for the researchers to develop a process called expert judgment. In this process, a brainstorming and round table discussion was implemented with the participation of major social actors in the study area, such as tourism directors, businessmen, tourist guides, academics, and political representatives.

**Table 1.** Methodological process for the bibliographic review.

| Topic | Search Parameters | Scopus/Web of Science |
|---|---|---|
| **Historical situation of the conflict and tourism** | ("Post-conflict" and "Tourism") and ("Colombia"/"Ecuador") | 7 [34–40] |
| **Current situation COVID-19** | ("COVID" and "Tourism") and ("Colombia"/"Ecuador") | 8 [41–48] |

| Historical situation of the conflict and tourism | | Current situation COVID-19 | |
|---|---|---|---|
| **Title** | DOI/URL | Title | DOI/URL |
| **Ecotourism in Colombian Peacebuilding: Peace, Conflict and Environmental Justice** | https://doi.org/10.4000/viatourism.4052 (accessed date 10 January 2021) | COVID-19 and dengue, co-epidemics in Ecuador and other countries in Latin America: Pushing strained health care systems over the edge | 10.1016/j.tmaid.2020.101656 (accessed date 12 January 2021) |
| **Social Entrepreneurship and Sustainable Tourism in Colombia: A Baseline Study in Post-conflict Regions** | https://doi.org/10.18848/2325-1115/CGP/v16i02/65-8 (accessed date 10 January 2021) | Management of digital strategic communication of the main companies in the tourist and gastronomic sector of Ecuador | https://n9.cl/ahgzs (accessed date 12 January 2021) |
| **Peacebuilding and post-conflict tourism: addressing structural violence in Colombia** | https://doi.org/10.1080/09669582.2020.1869242 (accessed date 10 January 2021) | Analysis of the measures taken by the Governments of Colombia and Ecuador in favor of the tourism industry during the pandemic generated by COVID-19 | https://n9.cl/qzllw (accessed date 12 January 2021) |
| **Colombia in post-conflict: Tourism for peace or peace for tourism?** | 10.12795/araucaria.2018.i39.20 (accessed date 10 January 2021) | Policy (in)capacity in COVID-19′s times: understanding the economic responses of Colombia and Ecuador | https://doi.org/10.15446/anpol.v33n100.93362 (accessed date 12 January 2021) |
| **Branding and promoting a country amidst a long-term conflict: The case of Colombia** | https://doi.org/10.1016/j.jdmm.2016.10.001 (accessed date 10 January 2021) | Situation of the Colombian Tourism Sector during the pandemic, a light at the end of the road: Lamentation or call to action? | https://n9.cl/3trm9 (accessed date 12 January 2021) |
| **Economic and Conservation Potential of Bird-Watching Tourism in Postconflict Colombia** | https://doi.org/10.1177/1940082917733862 (accessed date 10 January 2021) | The country brand in times of confinement: Analyzing the publicity message of tourism promotion of Spain and Colombia during COVID-19 | https://n9.cl/r56ql (accessed date 12 January 2021) |
| **Conflict, Environment and Transition: Colombia, Ecology and Tourism after Demobilisation** | https://doi.org/10.5204/ijcjsd.v8i3.1246 (accessed date 10 January 2021) | Perception of Safety Tourism in Colombia | 10.1007/978-981-33-4260-6_9 (accessed date 12 January 2021) |
| | | Community tourism in Ecuador: Notes in times of pandemic | https://n9.cl/57mh6 (accessed date12 January 2021) |

## 3. Results

The bibliographic review was carried out and added to the analysis through the judgment of experts in the Amazonian border of Colombia (Department of Putumayo) and Ecuador (Province of Sucumbíos). The results of the analysis and description of the historical situation of the armed conflict and post-conflict, the current situation focused on the COVID-19 pandemic crisis, and the challenges in the Amazonian border of Colombia (Department of Putumayo) and Ecuador (Province of Sucumbíos) with respect to nature tourism are presented below.

### 3.1. History of Conflict and Post-Conflict in Nature Tourism

The armed conflict in Colombia asted a little more than half a century until 2016, when the peace agreement was signed between the Colombian Government and the Revolutionary Armed Forces of Colombia (FARC) [49–51]. The conflict was considered the longest lasting and most complex to manage in the Western hemisphere. According to Colombian government data, the officially recognized victims amounted to 220,000 dead, and some seven million people left their original regions in search of better lives [34]. Recent publications describe that this conflict in the Amazon region greatly affected neighboring Ecuador, specifically the Province of Sucumbíos. FARC members used the border area between the Department of Putumayo and the Province of Sucumbíos to avoid controls and increase their illicit crop cultivation capacity; the dense vegetation and jungle area was conducive to hiding their camps and plantations [14,35]. Thus, this area was considered strategic during the half-century of armed conflict. All these events were known to potential tourists from European, Asian, and North American countries.

Publications in the last five years agree that tourism activities in the study area have been reduced and limited by the armed conflict since the 1980s and mainly in the 1990s. Nature tourism in the Department of Putumayo and the Province of Sucumbíos in those decades was considered emerging tourism, but its consolidation was conditioned by political instability [36,37,52]. Land access roads in the Amazon in those decades were in very poor condition (Figure 1a), and it was very common to observe armed personnel (Figure 1b) patrolling and carrying out controls, aspects that instilled generalized fear in the few visitors to the region [38,39]. In this research, 125 tourists of European nationality (74%: Germans 33%, Spanish 30%, French 7%, others 4%); and 26% from North America who visited the study area were interviewed. All (100%) of the interviewees were between 20 and 35 years of age, 69% were men and 31% were women. All of them stated that they had not come to the Amazonian area between Ecuador and Colombia before because they associated it with violence, as a result of armed guerrilla conflicts and drug trafficking. In addition, they indicated that in their countries' lists of red alert destinations not to be visited, the border zone of Ecuador and Colombia in the Amazon were the first to appear, along with countries in the Middle East.

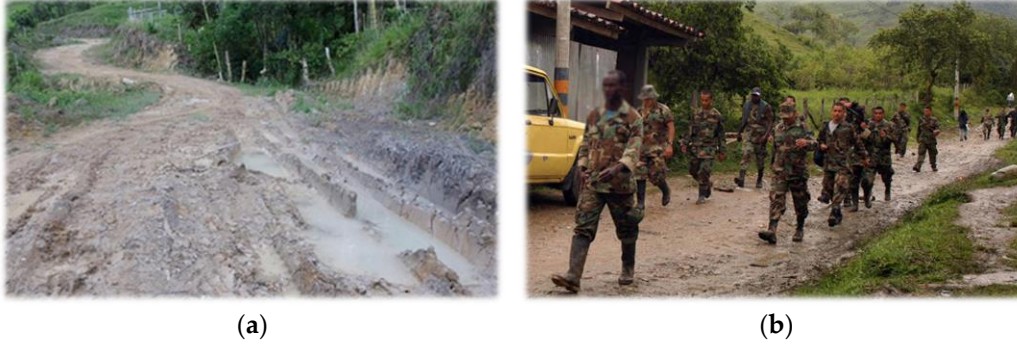

(a)         (b)

**Figure 1.** Historical problems in the Colombian–Ecuadorian Amazon. (**a**) Roads in poor condition, a problem of the last two decades of the 20th century. (**b**) Armed people traveling through Amazonian villages for unclear purposes.

The Cuyabeno Wildlife Reserve is considered one of the most important protected areas in the Ecuadorian Amazon because of its potential as a natural tourism resource [53]. Interviews with social actors in this protected area indicated that between 2000 and 2012, there were kidnappings and assaults in the river transit zones to access the reserve's public tourism zones. In February 2012, 20 tourists, including Europeans, Chileans, and Americans, were detained for approximately one hour, during which time their documents and personal belongings were stolen. Another case, considered one of the most important, occurred in September 2012, when two tourists from Australia and the United Kingdom were kidnapped while entering the Cuyabeno Reserve by canoe. These events, along with others, were associated at the time with conflicts between the FARC and the Colombian government. These acts set off alarm bells in European countries, North America, and Oceania. The governments of countries such as the United Kingdom, Canada, Australia, and Germany recommended their tourists to avoid visiting the Amazonian border of Colombia (Department of Putumayo) and Ecuador (Province of Sucumbíos) as far as possible because it was an insecure area with criminal activity including kidnapping. It was evident that these conflictive episodes in the first two decades of the 21st century affected nature tourism in the Amazonian area between Colombia and Ecuador.

The analysis of the panel of experts (researchers), together with the social actors of the region agreed that the main historical problem identified was the lack of knowledge of tourism policies, which has prevented the availability of a tourism development plan. Another of the problems shared in the Amazonian zone of Colombia and Ecuador is the absence of a structured offer of tourism products and services, which, added to the problems of insecurity due to armed groups, as well as the disorganization, disarticulation, and informality of tourism service providers, are aspects that have prevented the proper promotion of tourism to strengthen nature tourism. However, from 2016 to the present, in an era called post-conflict, nature tourism was presented as a great opportunity for socioeconomic growth.

A review of visitor statistics for the study area shows that the number of visitors has increased over the last decade. Authorities say that it used to be unthinkable to receive tourists in the region, but now that the perception of security is changing, more and more people are daring to visit the Amazon region [37]. The destination was gaining strength until before the COVID-19 pandemic, something very important for the region because the stigmatization was being left in the past and tourists began to look differently to this area in a post-conflict era, providing an opportunity for nature tourism [40,53]. Thus, in the Department of Putumayo in the last years of conflict, between 2012 and 2016, the number of foreign visitors increased by 67.13%, from 508 in 2012 to 849 in 2016. In the post-conflict years, the statistics for the Department were even better, between 2016 and 2019 the increase in tourists tripled, from 849 to 2503 visitors [54]. Likewise, Sucumbíos in Ecuador has presented historical changes; there is no record of visitors in its entirety as a province, but its most emblematic protected area, the Cuyabeno Wildlife Reserve, has been taken as a reference, where in 2011 it received 10,169 visitors, and by 2019 this number had almost doubled, reaching 17,404.

### 3.2. Current Situation and the Pandemic Crisis in Nature Tourism

All the documents analyzed agreed that since the 1970s until the 2010s, the Amazon region on the border of Ecuador (Sucumbíos) and Colombia (Putumayo) was perceived as a zone of conflict and violence. Today, after the peace process and great efforts of its authorities and communities, this area promises to be an emblematic destination for nature tourism [42,50,55,56]. In recent years, this region has prioritized tourism as a potential sector for income generation and competitiveness, to the extent that the advantages of being considered an area of great biodiversity and cultural wealth can be exploited [45,57,58]. This constitutes a great opportunity for local economies, and at the same time a challenge of trying to promote sustainable tourism and the respectfulness of nature, the environment, and host communities.

Reviewing tourism flow statistics in the new post-conflict stage, the Department of Putumayo was one of the destinations in Colombia with the highest growth in terms of foreign tourist arrivals [54]. On the other hand, because of the COVID-19 pandemic, foreign tourist arrivals decreased, so that in the period 2016–2019, after a growth of 194% since the signing of peace, by 2020 the presence of foreign visitors decreased to 762, a decrease of 66%, returning to numbers similar to the times of the conflict.

According to the panel of experts formed for this analysis, when the COVID-19 pandemic began, and in accordance with most media reports which indicated that biodiversity in general would show a recovery, in the study area it has become evident that this is not so true. Due to the closing of borders and the paralysis of tourism in the Colombian–Ecuadorian Amazon, there is evidence of a destructive effect on wildlife and the communities that protect them. Recently, poaching and looting have increased in protected natural areas and sites of tourist interest (Figure 2a). In turn, the natives, lacking economic income, have returned to hunting and fishing activities with procedures that affect biodiversity (Figure 2b), causing great disturbance. All of this is due to the fact that there are fewer tourists, and the numbers of control and monitoring personnel have been significantly reduced. Heritage conservation in the cultural sector has also been affected, as well as the cultural and social fabric of the communities, particularly in the case of indigenous peoples and ethnic groups. For example, the closure of markets for handicrafts, products, and other goods has especially affected the income of indigenous women. Another current problem is the postponement of intangible cultural heritage practices, such as traditional festivals and gatherings, which has had a significant impact on the social and cultural life of communities in the Amazon.

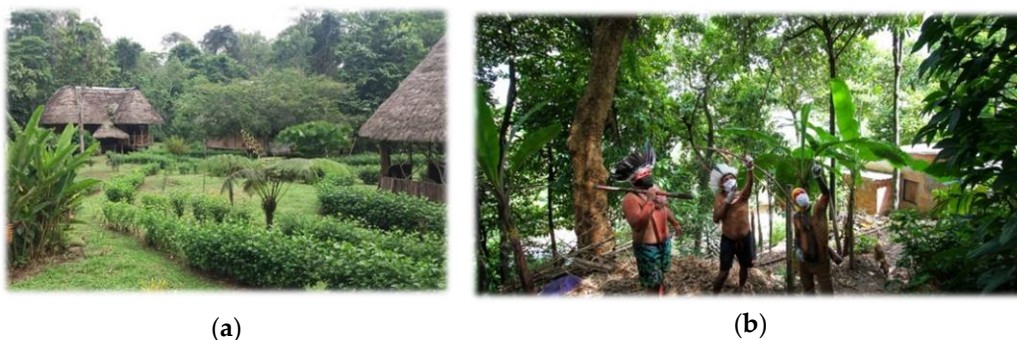

(**a**)                                        (**b**)

**Figure 2.** Pandemic effects on nature tourism. (**a**) Abandonment and deterioration of infrastructure in community-based tourism enterprises. (**b**) Natives returning to hunting activities.

It is clear that this pandemic has highlighted the interrelationship that exists between various stakeholders and sectors, including private enterprise, public health, visitors to natural sites, gateway communities, government, and non-governmental organizations. If we aspire to restore nature-based tourism in rural areas rich in biodiversity, with healthy ecosystems, resilient livelihoods, and sustainable tourism economies, recovery requires challenges where the collaboration and integrated work of tourism stakeholders is paramount.

### 3.3. Post-Conflict and Post-Pandemic Challenges in Nature Tourism

Tourism plays a crucial role in job creation, foreign exchange generation, and economic activity in general [59,60]. After a post-conflict stage in the Amazonian zone of the Colombia–Ecuador border, tourism showed signs of growth; however, this activity has come to a drastic halt worldwide as a result of the COVID-19 pandemic. Authorities in the study area should focus their efforts on reactivating the tourism economy in a post-pandemic scenario. The pandemic has changed consumer trends, and while activities have not been fully reactivated, it is possible to analyze what the main demands of the future will be. In the case of tourism, people will mainly look for natural, uncrowded places that

allow them to maintain a greater social distance, i.e., where there is less risk of contagion. Thus, tourists worldwide after the pandemic will prefer sustainable tourism (34.6%), with natural immersion (29.3%) and authentic local experiences (52.0%).

A post-conflict and post-COVID situation provides the opportunity to rethink new objectives for nature-based tourism and its contribution to people and the planet. It is a new opportunity that, in rebuilding it, the sector will be better, more sustainable, inclusive, and resilient, and that the benefits of tourism will be shared more widely and fairly. Due to its cross-cutting economic nature and deep social footprint, tourism is uniquely positioned to help affected societies and communities return to growth and stability. Over the years, the sector has consistently demonstrated its resilience and its ability to not only recover as a sector, but to lead a broader economic and social recovery. It is important that the authorities in the Amazon region between Colombia and Ecuador are interested in strengthening dialogue and international cooperation. This common challenge presents an opportunity to collaborate more closely and show that solidarity can transpire beyond national borders.

The problems of health contagions, with the presence of new waves and sepsis of COVID-19, coupled with a history of conflict and criminality in the Amazonian area on the border of Ecuador (Sucumbíos) and Colombia (Putumayo) is perhaps one of the greatest challenges for the sector and the governments. To face this challenge, not only are strict security and sanitary measures needed, but also strategies to manage and guarantee a manageable flow of visitors after a post-conflict and post-COVID process. Efforts must be made to improve visitors' experiences and sensations by incorporating adventure activities and taking advantage of the area's biotic, abiotic, cultural, and landscape resources (Figure 3). Providing a quality service that enhances experiences will generate an excellent reputation [61]. In this context, there is an opportunity to rethink tourism and erase the past fears of conflict and the current fears of a health crisis, promoting a new sustainable tourism model.

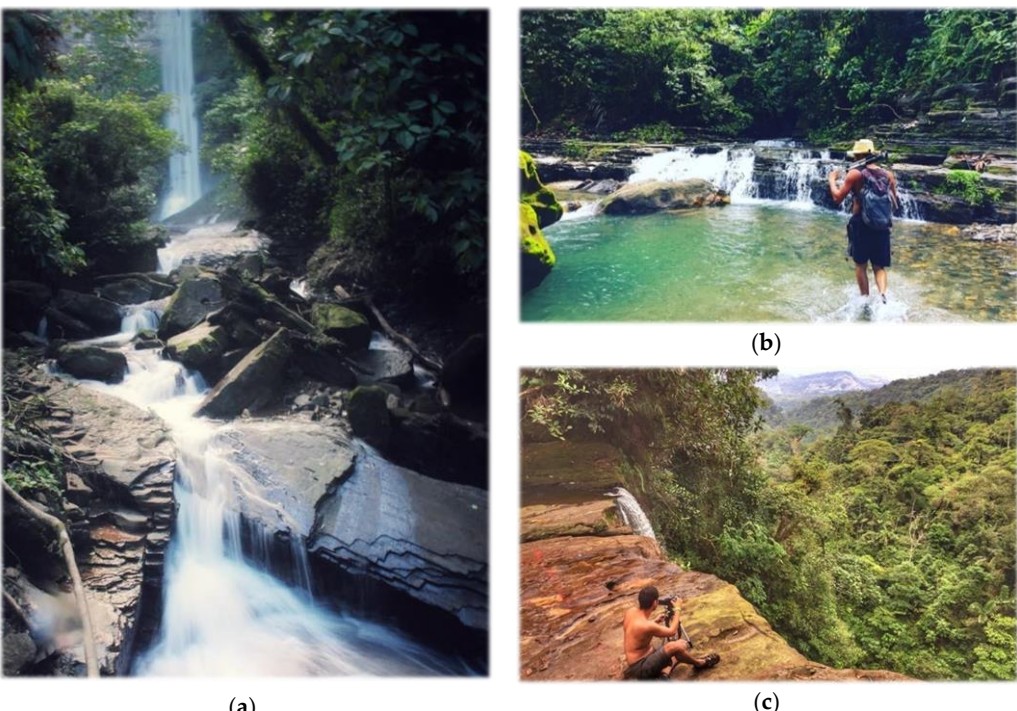

**Figure 3.** Natural tourism resources in the Colombian–Ecuadorian Amazon. (**a**,**b**) La Libertad waterfall, Sucumbíos, Ecuador. (**c**) Waterfall at the end of the world, Putumayo, Colombia.

It is important to focus efforts on the creation of new policies to take advantage of the potential of tourism resources in the Amazon region between Ecuador and Colombia in a

sustainable manner. Authorities must join efforts to incorporate technological advances to all processes in the nature tourism sector. Digitalization is no longer an option; it is a necessity. In a post-conflict and post-COVID period, new ways of working and understanding the needs and requirements of potential clients are required. The incorporation of new technologies is essential to create more and better jobs and make this tourism sector more operative and capable of responding to adversities. For this, it is necessary to strengthen professional training and create a change of mentality that will allow this sector of tourism to adapt to the new reality of the tourism industry.

This post-conflict and post-pandemic tourism scenario should become one of the pillars of public policy on the Amazonian border in the Department of Putumayo and the Province of Sucumbíos, Ecuador. These two jurisdictions should join efforts and create binational routes in the sector of nature tourism, which would enable actors to take advantage of the richness of its natural resources. A great challenge for all those involved in tourism is to think of a post-growth concept, in which the need arises to develop a model of society in which we have managed to overcome the idea of economic growth alone. It is time to focus on those aspects that effectively contribute to a better welfare where the fundamental objective is social, environmental, and cultural policies, which provide a real welfare and whose benefits are at least at the same level or integrated with the economic benefit that these could generate.

## 4. Discussion

The literature review has shown that political and health crises generate instability and a significant impact on the tourism industry [6,62,63], generating negative effects such as the decrease in visitors that has resulted in low economic income. There is evidence of cases that have gone through similar episodes and have never been able to recover, but there is also evidence of destinations that have applied good strategies and their recovery has been evident [32,40]. The study area shows that the recovery was on the right track, as evidenced by the visitor statistics in the post-conflict period [23,52]. Now that the post-conflict process is over, the only thing left to do is to continue applying existing strategies and implement new ones to provide tourists with security and confidence [64]. Studies show that a good promotion of avant-garde measures in natural areas will attract large groups of tourists who are currently interested in the type of destination that the Amazonian area on the border of Ecuador (Sucumbíos) and Colombia (Putumayo) has to offer.

The Colombian–Ecuadorian Amazonian border zone after five decades of conflict has shown that tourism can recover in the short term. It is important to highlight that prior to the signing of the peace agreement in 2016, the authorities were already working on strategies to strengthen tourism activities when the conflict ended [65]. This meant that there was no tourist hibernation process, and in only five years after the signing of the peace agreement, the number of tourists in the study area tripled. Thus, conflict-affected areas can recover quickly, depending on how prepared they are for the transition from chaos to peace. The joint effort of tourism managers in post-conflict areas should focus on harnessing the potential of tourism to promote peace. A good post-conflict tourism management will allow, in the short or medium term, areas to respond promptly in the creation of employment, dynamization of the economy, and redistribution of income [35,66,67]. On the other hand, the tourism mission and vision should be elevated to the level of a strategic sector of national development, as a state policy. This will make it possible to turn the tourism sector into a true path of sustained, sustainable, and proactive economic and social development, linked to the country's economic dynamics. The formation of a tourism community should be based on a common objective, allowing for the construction of a business/academic/professional fabric under the criteria of research and joint and participatory action. Taking as a reference successful tourism recovery processes in countries that have gone through similar situations is essential to avoid making strategic management mistakes.

The results show a significant drop in tourism activity in the Colombian–Ecuadorian Amazonian border area, as in the rest of the world [68]. The reactivation of tourism activities will play an important role for countries that depend heavily on this industry from an economic point of view [69]. Currently, the economic crisis has put pressure on governments to gradually reopen tourism activities and increase their resilience to adverse situations [70]. There is still a high risk of contagion from new outbreaks, and science is working tirelessly in search of an effective solution to combat the virus. It is important to establish increasingly effective strategies and measures to reactivate tourism after a global pandemic situation, aspects that will make it possible to provide greater security to visitors. It is evident that the perception of travel risk has changed after the pandemic crisis. Given this reality, it is important to know the current preferences of travelers and the aspects considered when choosing a destination and thus being able to meet the new needs.

It is evident that tourism is vulnerable to crisis or disaster situations. These events cause serious economic and environmental damage. However, there is also evidence that the high demand for tourism makes this industry very resilient to a crisis [32]. Currently, in a post-crisis situation, tourists are looking for new forms of tourism that will allow activities with a lower risk of contagion [62]. Given this, the Amazonian area on the border of Ecuador (Sucumbíos) and Colombia (Putumayo) enjoys an advantage and opportunity for a resurgence in terms of nature tourism. The issues of the conflict in the study area will always be present, because it is difficult to forget the past violence and terrorism, but there is great evidence that it is possible to change and to re-emerge from an adverse situation [29,71,72]. Thus, tourism activities contribute positively to socioeconomic development and the strengthening of peace among different actors. Adverse post-conflict and post-COVID events can be used strategically to stimulate information and knowledge tourism that narrates the strongest episodes of the war and its process of change, complemented with the natural wealth of the region: its main landscapes such as mountains, rivers, waterfalls, indigenous communities, lakes, and lagoons.

## 5. Conclusions

All tourist destinations and activities around the world are exposed to conflicts or health problems that are generally not predictable. Tourism can play a crucial role in social and economic changes, and by proposing future scenarios for the recovery of these places can develop in the short, medium, or long term. Everything will depend on the management and strategies of the social actors of the tourism industry in each destination or region. The main findings of the study are highlighted below.

The armed conflict since the 1970s until the 2010s affected the socioeconomic development in the Amazonian border area of Colombia and Ecuador. This situation impeded the economic investment of the governments to improve the quality of life of its population, affecting the development and emergence of tourism activities such as the creation of products or services to take advantage of the richness of its natural resources. In the second decade of the 21st century, the local governments of the Department of Putumayo in Colombia and the Province of Sucumbíos in Ecuador focused their efforts on economic investment to strengthen the development of nature tourism in areas that, for years, have been scenes of displacement, fighting, illicit crops, drug trafficking and territorial control.

On the other hand, the fear of COVID-19 has generated great uncertainty and chaotic conditions in the tourism industry, experiencing a sharp drop in income, being one of the economic sectors most affected by the pandemic. The impact affects both the demand side (restrictions on the freedom of movement, border closures, fear of infection among guests) and the supply side (closure of establishments and staff layoffs) in the study area. Tourism activity can be an instrument that involves the actors who were part of this post-conflict process and those who are currently part of a post-pandemic process, allowing them to improve their living conditions. The Amazon region on the border of Colombia (Department of Putumayo) and Ecuador (Province of Sucumbíos) currently has a unique and privileged opportunity in a post-conflict and post-COVID situation. Its correct use

and success will depend on the policies and strategies undertaken by the main tourism stakeholders. It is important to emphasize that a joint effort between the authorities and the private sector related to tourism is needed to build and offer quality products and services, with the security required by the current situation. NGOs and academia should play an important role, providing support and advice in achieving the objectives in this new era of tourism.

Furthermore, from a theoretical point of view, it is noteworthy that research on resilience has made significant progress in the last two decades. In this study, we try to contribute, as main theoretical implications, the understanding of new forms of resilience in post-conflict and post-pandemic areas, and to provide a debate on the relationship between tourism and the recovery or development of areas degraded by conflicts or pandemics. All of this should contribute to strengthening the body of disciplinary knowledge in this area of research.

Finally, the knowledge of the reality of nature tourism in the Colombian–Ecuadorian Amazon in a post-conflict and post-pandemic situation opens the way to new research that should focus on studying the potential of natural resources and their landscapes (rivers, waterfalls, lagoons, mountains and communities), which will allow the creation of quality tourism products and services through policies and strategies adapted to current tourism requirements. It is important to highlight that the main limitation of the study was the lack of scientific information available in high-impact databases that related tourism to conflict episodes in the study area. We hope that this contribution and the analysis of gray literature will strengthen the information available on the history, current situation, and challenges of nature tourism in the Amazonian border of the Department of Putumayo in Colombia and the Province of Sucumbíos in Ecuador.

**Author Contributions:** Conceptualization, C.M.-R. and J.L.J.-C.; methodology, C.M.-R.; validation, J.L.J.-C.; formal analysis, J.L.J.-C.; investigation, C.M.-R. and J.L.J.-C.; resources, C.M.-R.; data curation, C.M.-R. and J.L.J.-C.; writing—original draft preparation, C.M.-R.; writing—review and editing, C.M.-R. and J.L.J.-C.; visualization, C.M.-R.; supervision, J.L.J.-C.; project administration, C.M.-R. and J.L.J.-C.; funding acquisition, C.M.-R. Both authors have read and agreed to the published version of the manuscript.

**Funding:** This research was funded by Instituto Tecnológico Superior Oriente (Grant No. 34323674).

**Institutional Review Board Statement:** Not applicable.

**Informed Consent Statement:** Informed consent was obtained from all subjects involved in the study.

**Data Availability Statement:** Data sharing not applicable.

**Acknowledgments:** The authors are grateful for the financial support of GREEN AMAZON ECUADOR, Instituto Superior Tecnológico Universitario Oriente (ITSO) and Escuela Superior Politécnica de Chimborazo (ESPOCH). As lead author, C.M.-R., I thank the Doctoral School of the University of Seville for allowing me to pursue a doctoral.

**Conflicts of Interest:** The authors declare no conflict of interest.

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
