# Peer review of "Nature Tourism on the Colombian—Ecuadorian Amazonian Border: History, Current Situation, and Challenges"

_sustainability, doi:10.3390/su13084432_

Round 1
Reviewer 1 Report
Thank you for the opportunity to review this paper for Sustainability once again. This is an interesting article with a good topic on nature tourism. My comments on it are as follows:
Abstract
It is rare to see some acronyms without any notes here when they first appear in the Abstract.
Introduction
1.Please provide some clear definitions of nature tourism and define it with your own unique perspective.
- Lack of enough literature to support your points.
- It would be better for the author to explicate why post-conflict and post-covid were discussed in your article in terms of adverse situations in tourism and why other modalities are not.
Methodology
- What specific techniques did you employed in your research when you conduct bibliographic review analysis. For example, content analysis.
- Please insert a table which compromise the titles and DOIs of the documents used for the bibliographic review.
- What does gray literature refer to?
- Which exact techniques did you use during expert judgement, Delphi technique, brain storming, or other approaches?
Result
Could you provide demographic information of respondents in the study?
Conclusions
What are the theoretical implications in this paper?
Author Response
Dear reviewer, please find attached the details of the corrections applied according to the suggested observations. Please, if any of them have not been correctly interpreted by us, or in your opinion are incomplete, please do not hesitate to clarify our misinterpretation. Thank you for all your comments, this helped us to improve our manuscript.

Reviewer 2 Report
Thank you for the opportunity to read this manuscript, as it is very well structured and presents an interesting subject. However, several important aspects need to be improved. The review gives several reading recommendations on several relevant works which can be found on Google Scholar.
Regarding the development of cross-border tourism in mountain regions regarding nature tourism in a different institutional context, please see:
Stoffelen, A., et al. (2017). "Obstacles to achieving cross-border tourism governance: A multi-scalar approach focusing on the German-Czech borderlands." Annals of Tourism Research 64: 126-138.
Paunović, I. and V. Jovanović (2017). "Implementation of Sustainable Tourism in the German Alps: A Case Study." Sustainability 9(2 (226)): 1-15.
Comments:
Line 11: please correct sentence structure: “…suffering from conflicts, acts that can indirectly affect bordering nations”
Lines 11-17: In my opinion, key-word “resilience” is missing at this point for describing the need to have a contingency (risk management) plan for conflicts and pandemics. The abstract needs to be reformulated to be more precise in defining its purpose as dealing with post-conflict and post-covid as future scenarios. The way it is formulated now, this is not apparent and can be mistaken for a present or even past situation. Please fine-tune the syntax in this text part.
Lines 53-55: Regarding personal safety, it has been demonstrated in the literature below that reliability of police services is the 6th most relevant indicator distinguishing developed from developing destinations:
Paunovic, I., et al. (2020). "Developing a Competitive and Sustainable Destination of the Future: Clusters and Predictors of Successful National-Level Destination Governance across Destination Life-Cycle." Sustainability 12: 4066.
Lines 56-57 as well as 154-156 and 167-183: It is not only the overall destination image that suffers from an unsafe environment, but more specifically certain transport modes. Cost, travel time, comfort, flexibility, safety, as well as other factors influence travel mode decision-making, where travelers consider travel alternatives for travelling to a chosen destination. Please see:
Mamula Nikolić, T., et al. (2021). "Sustainable Travel Decision-Making of Europeans: Insights from a Household Survey." Sustainability 13(4).
Karl, M. (2016). "Risk and Uncertainty in Travel Decision-Making: Tourist and Destination Perspective." Journal of Travel Research 57(1): 129-146.
Line 95: Similar to the comment regarding the abstract, I think the reformulation to “in a future post-conflict and post-pandemic situation” is necessary in order to be more precise in describing this research goal.
Good luck with the changes!
Author Response

(The authors gave the same response as above.)

Round 2
Reviewer 2 Report
Dear authors,
Thank you for the provided changes. There is however one minor issue which needs to be addressed in the article. The discussion should be expanded to include more implications of the results for the global academic and practitioner community. Right now, the accent is more on references, which is why own, original text should be expanded. Here argumentation should be provided as to how the results extend current debates around tourism during and after armed conflicts, around Covid 19 as another form of challenge for tourism resilience, as well as future outlook for post-conflict and post-Covid 19 destinations.
Good luck!
Author Response
Dear Reviewer,
The minor corrections requested in the discussion section have been applied. Two paragraphs were increased considering our results.
Thank you for your comments.